# Relationship of Photosynthetic Activity of *Polygonum acuminatum* and *Ludwigia lagunae* with Physicochemical Aspects of Greywater in a Zero-Liquid Discharge System

Karen Takahashi [1], Gabriela Araújo [1], Vali Pott [2], Nídia Yoshida [3], Liana Lima [2], Anderson Caires [4] and Paula Paulo [1,*]

1   Faculty of Engineering, Architecture and Urbanism, and Geography, Federal University of Mato Grosso do Sul, Campo Grande 79090-900, MS, Brazil
2   Bioscience Institute, Federal University of Mato Grosso do Sul, Campo Grande 79090-900, MS, Brazil
3   Chemistry Institute, Federal University of Mato Grosso do Sul, Campo Grande 79090-900, MS, Brazil
4   Physics Institute, Federal University of Mato Grosso do Sul, Campo Grande 79090-900, MS, Brazil
*   Correspondence: paula.paulo@ufms.br

**Abstract:** Landscape harmony is a key factor in the application of nature-based solutions to provide green areas. The search for plants that meet this requirement is crucial in this context. We evaluated the adaptation, resistance, and performance of *Polygonum acuminatum* and *Ludwigia lagunae*, macrophytes from the Pantanal biome, in greywater-fed mesocosms simulating zero-liquid discharge systems. Four irrigation solutions were tested for 212 d. Neither species exhibited stress conditions in the adaptation phase, with photosynthetic activity (*Fv/Fm*) close to that obtained in Pantanal. However, over time, the mesocosms irrigated with greywater (GW) without nutrient supplementation exhibited stress according to correlation analyses of photosystem PSII and physicochemical parameters; *L. lagunae* for dissolved oxygen below 3 mg $L^{-1}$ and *P. acuminatum* for water temperatures above 27 °C. Supplementation of GW with nutrients resulted in good growth and performance. Both species were able to receive high chemical oxygen demand (COD) loads, averaging 34 g $m^{-2}$ $day^{-1}$ for *L. lagunae* and 11 g $m^{-2}$ $day^{-1}$ for *P. acuminatum*, with an average removal of 85% by both. *L. lagunae* had better evapotranspiration capacity, with greater potential for use in cooling islands, whereas *P. acuminatum* showed a more resistant metabolism without nutrient supplementation.

**Keywords:** constructed wetlands; domestic sewage; heat islands; photosynthetic activity; water reuse

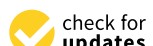



## 1. Introduction

Greywater (GW) is the fraction of domestic wastewater generated during personal and home care activities (namely, laundries, sinks, showers, toilets, kitchens, and dishwashers) but excluding water from toilets, and can account for about 85% of domestic wastewater [1]. It is a potential resource for reuse, as following low-resource-intensive treatment, greywater can be recycled for non-potable purposes (toilet flushing, irrigation, landscaping, and recharging groundwater aquifers).

The use of nature-based solutions (NBSs) to treat GW in urban areas can convert a significant fraction of wastewater into a water source to mitigate the impact of urban heat islands, flooding, and food supply, offering various ecosystem services beneficial to the environment [2–4]. NBSs are techniques that mimic natural processes in urban landscapes with low inputs of energy and chemicals [2]. Among the different NBSs, constructed (or treatment) wetlands (CW) have been studied and applied since the 1950s and have become a consolidated and reliable technology [5], widely applied for decentralised greywater treatment [4,6].

Plants may perform several roles in CWs, but their direct roles can be underestimated [7]. A recent review [8] summarised the most direct roles of the direct uptake of

nutrients, such as nitrogen, phosphorous, and sulphur, and aeration by plant roots. Plants, such as aquatic macrophytes, are potential options for CW applications because of their ability to grow in wetlands and provide structures from their root systems that assist in pollutant removal processes throughout aquatic macrophyte metabolism [9–11]. Adaptation to greywater is of concern, as greywater might be rich in diluted chemicals used for personal and house care [2,12]. Ornamental plants are usually utilised for secondary or tertiary treatments because of the reported toxic effects of high organic/inorganic loading on plants in systems that use them for primary treatment [13], whereas our research group is pursuing the use of ornamental plants in CW units without primary treatment. By finding proper species, both resistant and ornamental, it is more probable to increase the availability of cooling islands in urban areas by treating GW in decentralised CWs.

According to a literature review from Sandoval et al. (2019) [13], a literature survey of 87 CWs from 21 countries showed that the four most commonly used flowering ornamental vegetation genera were *Canna*, *Iris*, *Heliconia* and *Zantedeschia*. Looking for resistant plants with an ornamental appeal, our research group has already tested *Heliconia psittacorum*, *Cyperus isocladus*, *Arundina bambusifolia*, *Alpinia purpurata*, *Canna x generalis*, *Equisetum giganteum*, *Caladium Hortulanum* and *Hedychium coronarium* [6,14,15]. Among them, *Canna* × *generalis* is the most promising in terms of adaptation, growth, and survival. Recently, we started a trial with plant species from the Brazilian Pantanal [16,17], which is recognised worldwide as an important ecosystem representing the largest floodplain on the planet [18] and which accounts for a rich diversity of macrophytes with at least 280 identified species [19]. In a preliminary study, *P. acuminatum*, *L. lagunae*, and *Sesbania virgata* presented some bioactive compounds with antimicrobial properties (unpublished data), where the first the species showed more promising results, deserving further investigation. Therefore, the objective of this study was to assess the use of *P. acuminatum* and *L. lagunae* in greywater-fed mesocosms simulating zero-liquid discharge systems, focusing on the adaptation and resistance of these two species, in addition to their performance in greywater treatment.

## 2. Materials and Methods

### 2.1. Plant Species

Aquatic macrophyte seedlings *Polygonum acuminatum* Kunth (Polygonaceae) and *Ludwigia lagunae* (Morong) H. Hara (Onagraceae) were collected in December 2017 in the South Pantanal biome (19°34′37″ S and 57°00′42″ W). Registration number in the Brazilian national genetic patrimony management system: A60E575. Seedlings were cultivated in a greenhouse (20°30′09″ S and 54°36′44″ W) in Campo Grande, Mato Grosso do Sul, Brazil. The aquatic plants were acclimatised in the greenhouse, and the species were cultivated in 12 L volume pots filled with fine gravel and irrigated with tap water.

#### 2.1.1. Irrigation Solutions

The experiment was carried out using four types of irrigation solutions: tap water (TW); tap water plus nutrients (TW*); laundry greywater ($GW_L$) and laundry greywater plus nutrients ($GW_L$*). A modified Hoagland solution was used when the irrigation solution was provided with nutrient [20]. Eleven stock solutions (SS) were prepared and different volumes were added to the irrigation solution when required. The volume added of each solution and their chemical composition (g $L^{-1}$) were as follows: SS-A (5 mL $L^{-1}$) Ca(NO$_3$)$_2$·4H$_2$O: 236; SS-B (5 mL $L^{-1}$) KNO$_3$: 101; SS-C (2 mL $L^{-1}$) MgSO$_4$·7H$_2$O: 246.5; SS-D (1 mL $L^{-1}$) KH$_2$PO$_4$: 136; SS-E (1 mL $L^{-1}$) EDTA: 13.0; SS-F (1 mL $L^{-1}$) FeCl$_3$·6H$_2$O: 7.80; SS-G (1 mL $L^{-1}$) MnCl$_2$·4H$_2$O: 1.810; SS-H (1 mL $L^{-1}$) H$_3$BO$_3$: 2.860; SS-I (1 mL $L^{-1}$) ZnSO$_4$·7H$_2$O: 0.220; SS-J (1 mL $L^{-1}$) CuSO$_4$·5H$_2$O: 0.080 and; SS-K (1 mL $L^{-1}$) H$_2$MoO$_4$: 0.020.

To provide a greywater with more stable characteristics throughout the experiment, the $GW_L$ was produced in a laundry machine (WD9102RNW/XAZ Samsung 10.4 kg model) always washing two men's blue jeans and four women's blue jeans in room-temperature

water, using 60 mL of concentrated laundry detergent and 80 mL of softener, using the program type "cotton". The volume produced per batch was around 85.5 L.

### 2.1.2. Experimental Set-Up

After the acclimatization period, the seedlings were individually planted in non-transparent plastic pots of 21 cm height and top diameter and a total volume of 5.5 L. The pots were filled with 6 kg of fine gravel (porosity 0.44; particle size $d_{10} = 13$ mm, $d_{30} = 11$ mm, and $d_{60} = 10$ mm). A polyvinyl chloride tub of 20 mm diameter and 20 cm height was installed 5 cm from the edge. The tube (Sampling port 1—SP1) was used as a piezometer for measuring parameters using a probe and was also used as an irrigation water inlet point. In addition, a hose was connected to the bottom of the pot and extended externally to surface height (Figure 1) for determining evapotranspiration (for 1.5 L mesocosm$^{-1}$), and collecting effluent samples (Sampling port 2—SP2). The pots were randomly arranged and distributed into groups of four pots.

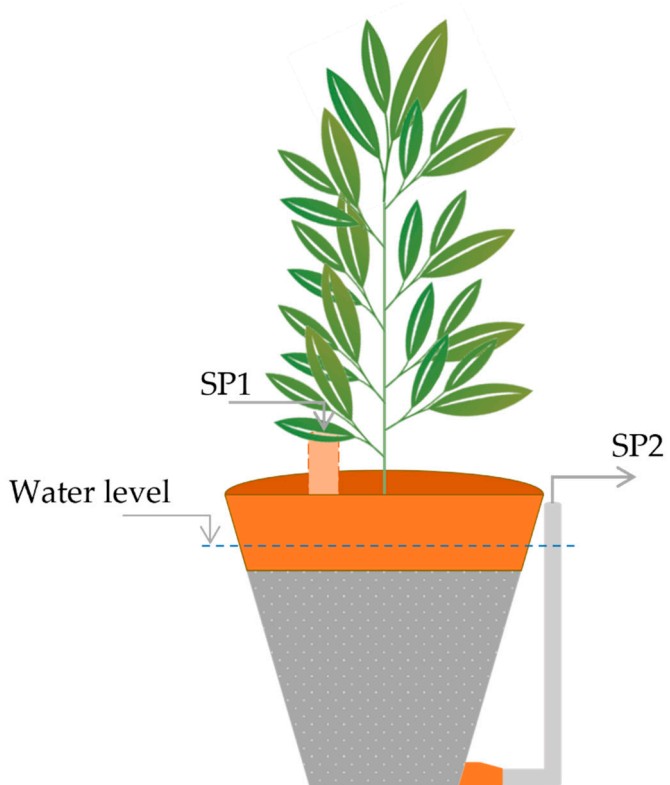

**Figure 1.** Schematic drawing of the mesocosms. SP1: sampling port 1, piezometer located inside the mesocosm; SP2: sampling port 2, located in an external hose used to keep the mesocosm liquid level, for emptying and for sampling.

The experiment was conducted in a greenhouse, situated at the university campus, to provide a controlled environment free of external influences such as wind and artificial lighting. A total of 24 mesocosms were used, distributed in triplicate for each type of irrigation and plant. The experiment was conducted in two phases: the experimental phase totalled 212 days, with the adaptation phase occurring in the first 35 days. The adaptation phase allowed the plants to adapt to the greywater, by its gradual addition. Initially, 1.5 L of TW were added to all the mesocosms. Every 7 days, the liquid medium was depleted, and new irrigation water was added. At each water renewal, the type of irrigation water (Table 1), diluted in tap water, was gradually added, with the concentration increasing by increments of 25% each week. The mesocosms in the control group received 100% TW at every change.

**Table 1.** Composition of irrigation solutions used during Stage 1 of the experiment.

| Experimental Day | Irrigation Solution | | | |
| --- | --- | --- | --- | --- |
| | TW | TW* | GW$_L$ | GW$_L$* |
| Day 0 (0%) | 100% TW | 100% TW | 100% TW | 100% TW |
| Day 7 (25%) | 100% TW | 25% TW* + 75% TW | 25% GW$_L$ + 75% TW | 25% GW$_L$* + 75% TW |
| Day 14 (50%) | 100% TW | 50% TW* + 50% TW | 50% GW$_L$ + 50% TW | 50% GW$_L$* + 50% TW |
| Day 21 (75%) | 100% TW | 75% TW* + 25% TW | 75% GW$_L$ + 25% TW | 75% GW$_L$* + 25% TW |
| Day 28 (100%) | 100% TW | 100% TW* | 100% GW$_L$ | 100% GW$_L$* |

TW: tap water; TW*: tap water + nutrient; GW$_L$: laundry greywater; GW$_L$*: laundry greywater + nutrient.

After the adaptation phase, irrigation was continued once a week, always after sample collection, with the addition of sufficient irrigation solution to make up the 1.5 L volume. Throughout the experiment, especially for plants that received nutrients, it was necessary to add irrigation water more frequently.

### 2.2. Measurements of Plant Growth

Non-destructive measures of the height and width of stems and the number of leaves were obtained for plants irrigated with the different types of irrigation solutions (TW, TW*, GW$_L$, and GW$_L$*). The height and width (neck) of the stems were measured using a pachymeter. The leaves considered for counting had a length of more than 7 mm. The measurements started one month before the experiment and were carried out until the 7th day after the addition of the last concentration (35th day) of the adaptation phase, with measurements taken once a week.

### 2.3. Photosynthetic Activity

The performance of the plants was evaluated using photosynthesis analysis. First, the photosynthetic metabolic pathway was determined to identify the metabolic type. Slides with transversally cut leaves of *P. acuminatum* and *L. lagunae* were prepared and submitted for analysis of leaf anatomy using a 100 μm lens electron microscope by the Botany/Plant Anatomy Laboratory of the Federal University of Mato Grosso do Sul. Fluorescence analysis was performed in vivo using a portable fluorometer (FluorPen FP 100, Photo Systems Instruments, Drásov, Czech Republic) to assess the photosynthetic activity of photosystem II (PSII) and chlorophyll a. Fluorescence analysis can determine the photosynthetic performance of plants in a non-destructive manner. Measurements were made before starting the experiments (day 0) to evaluate the behaviour of the species under natural conditions in the greenhouse. After day zero, photosynthetic activity was measured daily during the first 49 days of experiment and, subsequently, measurements were made daily except during weekends. In addition, triplicate measurements were made on *P. acuminatum* and *L. lagunae* on site in the Pantanal, in the same region where the seedlings used in the experiment had been collected.

Daily readings of the photosynthetic analysis were performed between 9:00 a.m. and 11:00 a.m., the time of the highest photosynthetic rates of the plants. The most visually healthy and fully developed leaves were selected for analysis. Measurements were always made on a single leaf, changing only when aging was observed or when the leaf had fallen.

The leaves were subjected to dark adaptation for 20 min prior to fluorescence measurements. Then, the initial fluorescence ($F_0$) was determined using low-intensity modulated light (less than 1 μmol m$^{-2}$ s$^{-1}$) for 50 μs. The maximum fluorescence ($F_m$) was determined with a pulse of saturating light (6000 μmol m$^{-2}$ s$^{-1}$) of 0.8 s [21,22]. The $Fv/Fm$ ratio was obtained from $F_0$ and $F_m$ ($F_v = F_m - F_0$); the ratio represents an important plant physiological parameter: the maximum quantum yield of PSII [23,24].

*2.4. Physicochemical Monitoring*

Measurements were performed weekly in two sampling stages. In the piezometer of sampling port 1, data on pH, temperature, dissolved oxygen (DO), redox potential (ORP), electrical conductivity (EC), salinity (SAL), and total dissolved solids (TDS) were analysed on site using a Hanna Edge® HI2002 for pH and temperature, a Hanna Edge® HI2040 for DO, an ORP meter ICEL OR-2300, and a Hanna Instruments HI2300 ET/TDS/NaCl meter (Hanna Instruments, Smithfield, RI, USA). For the second sampling stage, 200 mL samples were collected after discarding the first 30 mL to analyse chemical oxygen demand (COD) and total solids (TS). All analyses were performed according to standard methods for the examination of water and wastewater [25].

*2.5. Statistical Analysis*

The experimental data (plant growth measurements and photosynthetic activity ($Fv/Fm$) from the adaptation phase were statistically analysed to evaluate possible differences in the vigour of plants irrigated with different types of irrigation solutions. Additionally, the COD and TS data were analysed to determine the differences in their removal by species and types of irrigation. For this, normal distribution was verified, and variance analysis (ANOVA) followed by Tukey post hoc tests at 5% ($p < 0.05$) were applied. Pearson's correlations between $Fv/Fm$ and physicochemical parameters measured for irrigation water (inlet) and inside the mesocosms (SP1) were calculated using Minitab® version 18.1.

**3. Results**

*3.1. Plant Performance with Greywater Irrigation*

3.1.1. Plant Growth

Growth of the aquatic macrophyte *P. acuminatum* (Figure 2a–c) did not differ among the different types of irrigation water, and the greywater did not interfere with growth. In addition, nutrient supplementation made no difference to plant growth in terms of neck diameter, height, and numbers of leaves.

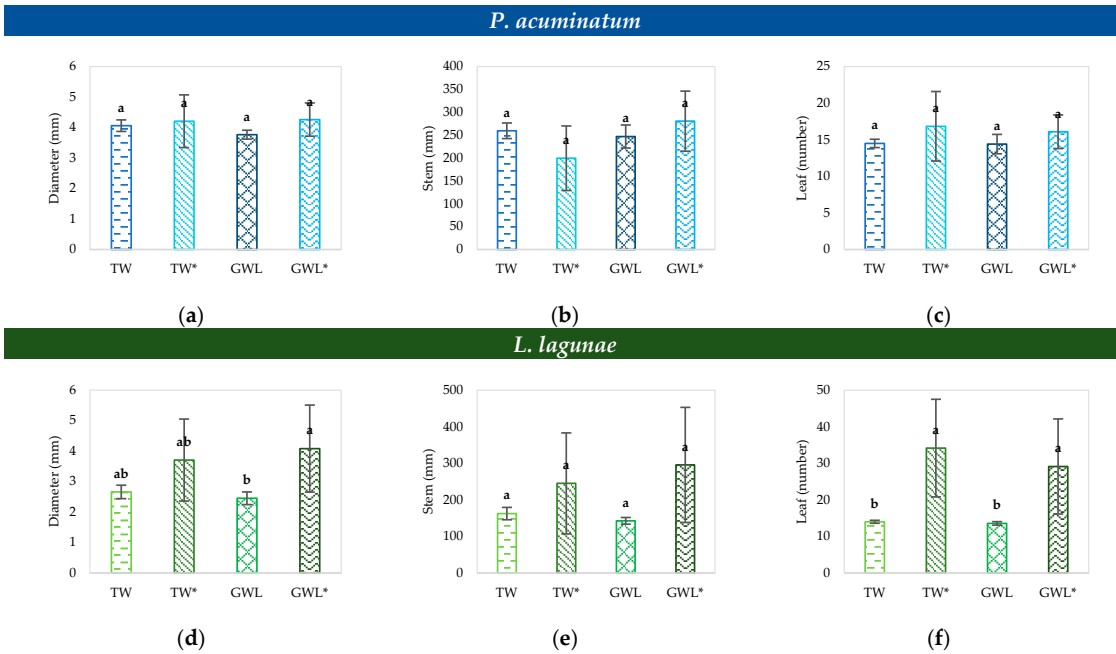

**Figure 2.** Mean growth of *P. acuminatum* (**a**–**c**) and *L. lagunae* (**d**–**f**) plants. Data: (**a**,**d**) neck diameter (mm); (**b**,**e**) stem height (mm); and (**c**,**f**) number of leaves. TW: tap water; TW*: tap water + nutrients; GW$_L$: greywater; and GW$_L$*: greywater + nutrients. The lowercase letters indicate statistical groupings based on ANOVA and Tukey tests. Values followed by the same lowercase letter are not significantly different at $p < 0.05$. Bars indicate the standard error of the mean.

In *L. lagunae*, the neck (Figure 2d) was greater in plants irrigated with greywater with nutrient supplementation and did not differ from the diameters of plants irrigated with TW and TW*. The number of leaves of *L. lagunae* plants irrigated with TW* and $GW_L$* was greater than that of plants where no nutrients were added to the irrigation solution (Figure 2e).

The results indicated that the nutrients were important for the diameter and number of leaves of *L. lagunae*; they were, respectively, 59% and (at least) 40% higher than those of plants irrigated with tap water without the addition of nutrients. The evapotranspiration of *P. acuminatum* and *L. lagunae* reached maximum values in the mesocosms irrigated with TW* and $GW_L$*, reaching 0.69 (TW*) and 0.56 ($GW_L$*) L day$^{-1}$ for *P. acuminatum* and 1.32 (TW*) and 1.33 ($GW_L$*) L day$^{-1}$ for *L. lagunae*. These values were reached between the 117th and 160th days of the experiment (Figure 3) when a high mean maximum temperature of 39.9 °C and a minimum temperature of 22.3 °C were reached in the greenhouse, with high relative humidity (93.6%).

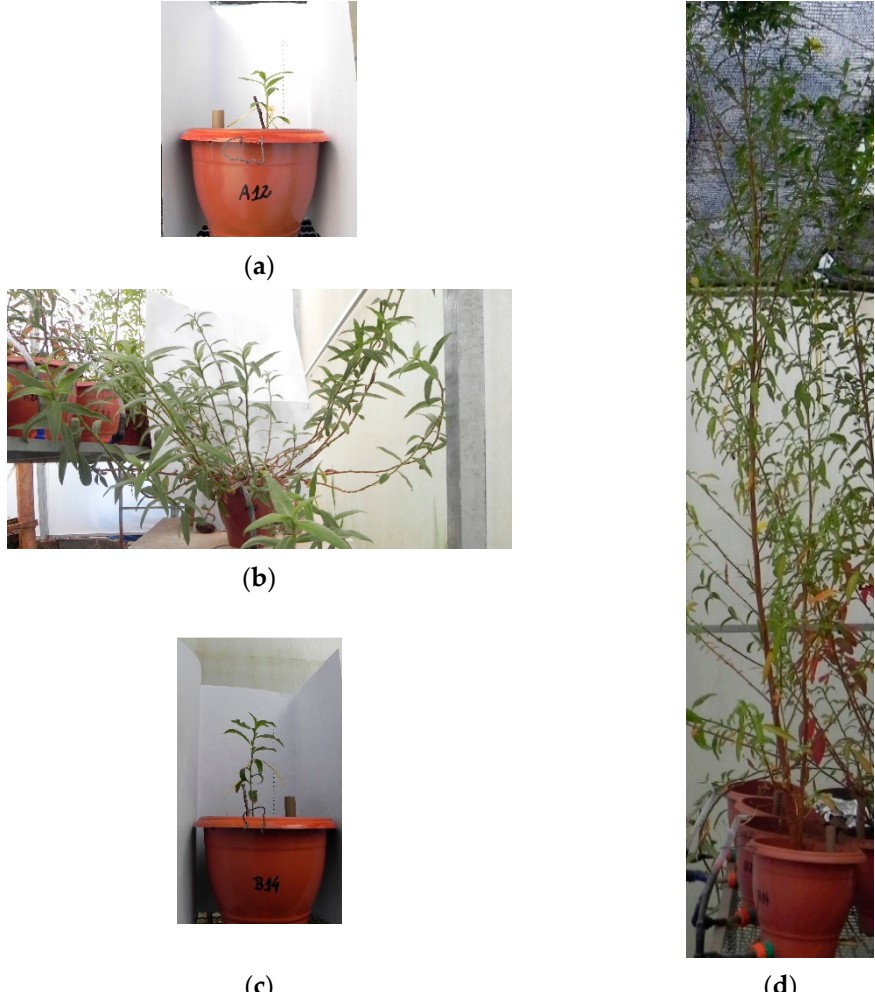

**Figure 3.** Images of the plant species at the beginning of the experiment, day 7 (**a**,**c**), and day 140 (**b**,**d**). Images (**a**,**c**) refer to *P. acuminatum*, and images b and d refer to *L. lagunae*, both irrigated with TW*.

3.1.2. Photosynthetic Activity

The leaves of *P. acuminatum* (Figure 4a) and *L. lagunae* (Figure 4b) did not show the anatomy of Kranz; therefore, they were classified as C3 photosynthesis type [26].

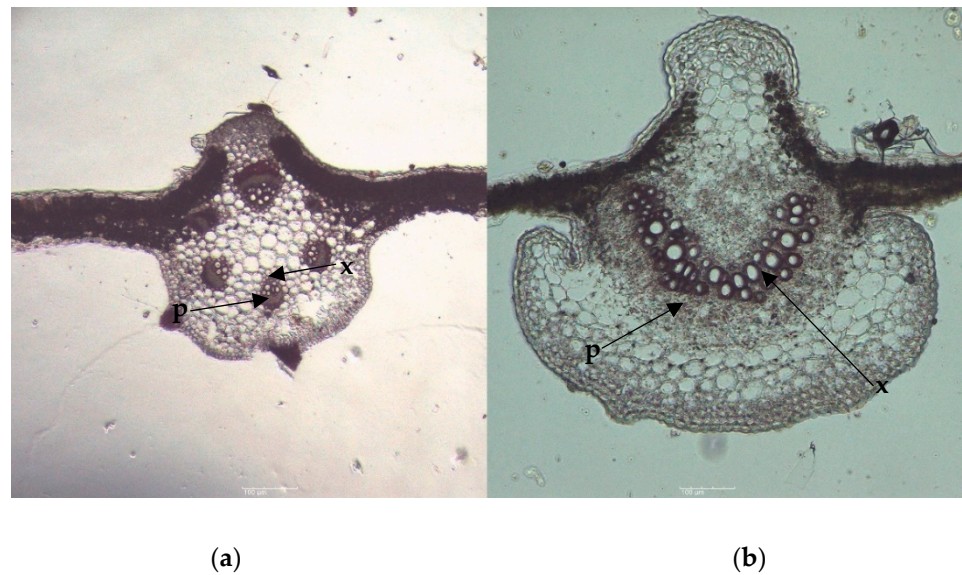

(**a**)          (**b**)

**Figure 4.** Microscopic image of leaf transverse section of *P. acuminatum* (**a**) and *L. lagunae* (**b**), with indication of the xylem (x) and phloem (p).

The quantum efficiency of the PSII system (*Fv/Fm*) for *P. acuminatum* during the first 51 d was constant for all treatments, with *Fv/Fm* values of approximately 0.80. Although the average *Fv/Fm* during this period was below the average value obtained in situ in the Pantanal biome (0.84 ± 0.01; Table 2), a normal function of PSII was indicated. After this period, a reduction in *Fv/Fm* was noted, indicating that the decrease in *Fv/Fm* over time was dependent on irrigation type (Figure 5a). A reduction in *Fv/Fm* was first detected in the $GW_L$ treatment (day 54), followed by detection in the control group (TW) on day 169. It took a little longer for a drop in *Fv/Fm* to appear in the treatments with extra nutrients supplied; the drop was observed in the TW* treatment on day 184, and in the $GW_L$* treatment on day 202.

**Table 2.** Average values for photosynthetic activity measured as *Fv/Fm* of *P. acuminatum* and *L. lagunae* plants irrigated with tap water (TW), tap water + nutrients (TW*), laundry greywater ($GW_L$), and laundry greywater + nutrients ($GW_L$*) during the first 51 days of the experiment (number between brackets are the standard deviation of at least three replicates).

| Plant Specie | Natural Habitat (Pantanal) | Day 0 | Tap Water (TW) | Tap Water + Nutrients (TW*) | Greywater ($GW_L$) | Greywater + Nutrients ($GW_L$*) |
|---|---|---|---|---|---|---|
| *P. acuminatum* | 0.84 ± 0.01 | 0.80 | 0.77 ± 0.06 | 0.79 ± 0.04 | 0.79 ± 0.02 | 0.81 ± 0.02 |
| *L. lagunae* | 0.83 ± 0.01 | 0.80 | 0.72 ± 0.04 | 0.78 ± 0.05 | 0.65 ± 0.12 | 0.78 ± 0.04 |

At the end of the experiment, the lowest photosynthetic activity was observed in plants irrigated with $GW_L$ only, with the drop in *Fv/Fm* reaching 0.28. This indicated that irrigation with $GW_L$ in zero-discharge systems was phytotoxic to the functioning of the photosystem and hence energy production, and the plant could not recover, indicating that this was due to the lack of nutrients. The same condition of irrigation with the addition of nutrients ($GW_L$*) was the one with the best performance, confirming the need for additional nutrients for the adaptation of plants to the wetland environment with zero discharge.

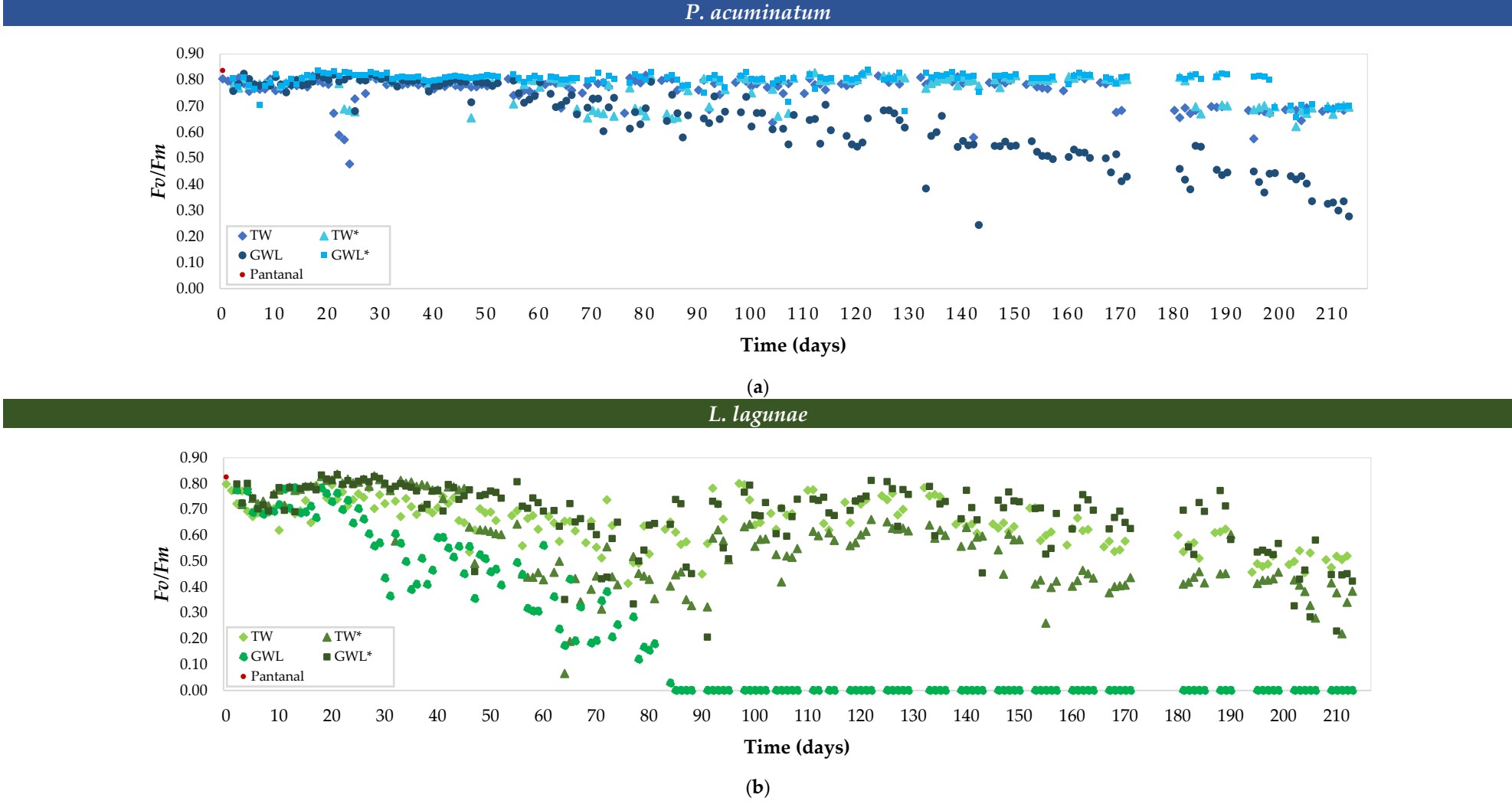

**Figure 5.** *Fv/Fm* over 212 days of experiments for *P. acuminatum* (**a**) and *L. lagunae* (**b**). Irrigation with TW (tap water—control); TW* (tap water + nutrients); $GW_L$ (laundry greywater); and $GW_L$* (laundry greywater + nutrients).

The quantum efficiency of the PSII system of *L. lagunae* in the control and nutrient-irrigated groups showed similar behaviours throughout the experiment (Figure 5b). Two periods were observed, each with an approximate duration of 90 days. The beginning of each period took approximately 50 days when high values of *Fv/Fm* were observed, followed by a fall. The first 51 days of the experiment (Table 2) showed that the three treatments with nutrients and the control had average *Fv/Fm* values below those obtained in-situ in the Pantanal biome. Nevertheless, the conditions with nutrients presented normal functioning of PSII.

Phytotoxicity to PSII was also observed in *L. lagunae* irrigated with greywater without nutrient supplementation. During the first 51 days, the lowest *Fv/Fm* was that of the $GW_L$ group, with an average of $0.73 \pm 0.04$, starting to drop on day 22, and reducing over time until reaching zero on day 84. When comparing the photosystems of the groups irrigated with $GW_L$ and $GW_L*$, a difference was clear, indicating that when nutrients were added, irrigation was sufficient, and even showed better responses than in the control and TW*.

### 3.1.3. Photosynthetic Activity and Physicochemical Characteristics

During the 212 days of the experiment, physicochemical analyses were carried out for samples taken inside the mesocosms (piezometer, SP1) and in the outlet (SP2) every time a new batch of irrigation water was added. Table 3 shows the average values of the analysed parameters for the different irrigation water types.

**Table 3.** Median values with standard deviation for the different solutions used to irrigate the mesocosms (the number of analysed samples are between brackets).

| Parameter | TW | TW* | $GW_L$ | $GW_L*$ |
|---|---|---|---|---|
| pH | $6.69 \pm 0.62$ (70) | $5.63 \pm 0.38$ (20) | $6.67 \pm 0.58$ (77) | $5.87 \pm 0.48$ (16) |
| Temp (°C) | $26.73 \pm 3.07$ (70) | $24.26 \pm 3.96$ (10) | $22.44 \pm 4.48$ (77) | $24.39 \pm 2.80$ (16) |
| EC ($\mu S\ cm^{-1}$) | $56.15 \pm 8.75$ (66) | $2636.35 \pm 769.55$ (20) | $179.92 \pm 30.49$ (62) | $2109.40 \pm 1084.13$ (16) |
| ORP (mV) | $349.53 \pm 178.36$ (65) | $431.00 \pm 88.11$ (19) | $227.04 \pm 173.64$ (77) | $318.87 \pm 27.49$ (15) |
| DO ($mg\ L^{-1}$) | $6.12 \pm 0.67$ (70) | $6.59 \pm 0.73$ (19) | $4.93 \pm 1.44$ (76) | $5.78 \pm 0.70$ (16) |
| COD ($mg\ L^{-1}$) | $9.43 \pm 3.21$ (6) | $19.87 \pm 13.74$ (14) | $325.72 \pm 71.37$ (37) | $342.79 \pm 116.12$ (16) |
| TS ($mg\ L^{-1}$) | $-^{na}$ | $1926.71 \pm 413.28$ (14) | $240.00 \pm 75.09$ (28) | $1940.77 \pm 718.02$ (13) |

TW: tap water; TW*: tap water + nutrients; $GW_L$: laundry greywater; $GW_L$: laundry greywater + nutrients; Temp: temperature; EC: electrical conductivity; ORP: oxy-reduction potential; DO: dissolved oxygen; COD: chemical oxygen demand; TS: total solids. $^{na}$—not analysed.

To better understand the influence of the variables studied, correlation analysis was performed, and only values of Spearman's $\rho$ that were above 0.70 were considered. A table with the correlation values is presented in the Supplementary Materials (Table S1). Data of *L. lagunae* irrigated with $GW_L$ that had null *Fv/Fm* were not taken into account in the correlation analysis.

The most significant correlation with *Fv/Fm* was obtained for ORP for *P. acuminatum* irrigated with $GW_L$ (Spearman's $\rho$ 0.80). Initially, the ORP was positive (Figure 6a) and above 300 mV until day 61, when the *Fv/Fm* was above 0.75, and the oxy-reduction potential decreased to values below 300 mV; the *Fv/Fm* decreased accordingly. This drop could be related to the increase of the temperature of the $GW_L$ irrigation solution (Spearman's $\rho$ −0.71), which began to increase on day 117, from $21.45 \pm 2.20$ °C, and continued to $27.56 \pm 2.28$ °C, when the ORP was below 200 mV and kept on decreasing, reaching −288.67 mV. From day 117 onwards, the temperature of all irrigation solutions increased owing to the temperature in the greenhouse, where the average minimum and maximum temperatures increased by 5 °C. Even with the non-significant correlation between DO and *Fv/Fm* of the mesocosms irrigated with $GW_L$ (Figure 6b), a decrease in DO was observed (Figure 6c) when the ORP reached negative values (146th day). Until then, the mean DO values were around $4.21 \pm 1.13$ mg $L^{-1}$, suffering a decrease around 50%, making the recovery of the photosynthesis of the plant difficult.

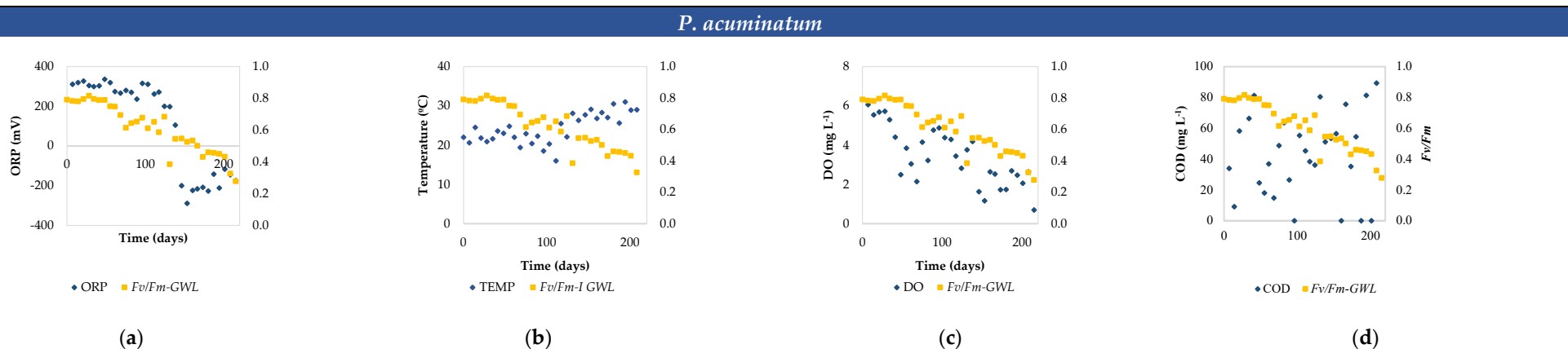

**Figure 6.** Relationships of *Fv/Fm* of mesocosms planted with *P. acuminatum* irrigated with laundry greywater without nutrient supplementation (GW$_L$), with (**a**) ORP; (**b**) temperature; (**c**) DO and; (**d**) COD. ORP: oxi-reduction potential; DO: dissolved oxygen; and, COD: chemical oxygen demand. Samples collected at the piezometers (SP1) except for (**b**) where I: irrigation solution.

For *L. lagunae*, the EC and salinity parameters showed an influence on the drop in *Fv/Fm* of plants irrigated with GW$_L$ (Figure 7a), and the most significant correlation was with EC (−0.85).

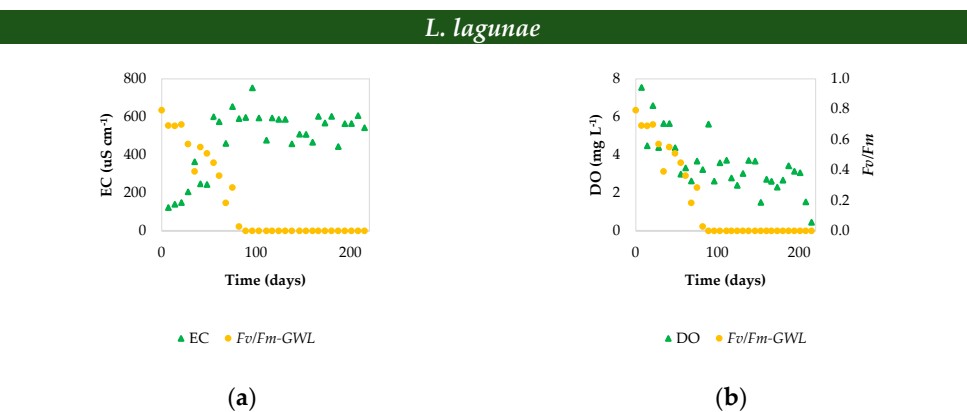

**Figure 7.** Relationships of *Fv/Fm* of mesocosms planted with *L. lagunae* then irrigated with laundry greywater without nutrient supplementation (GW$_L$), with (**a**) electrical conductivity (EC) and (**b**) dissolved oxygen (DO). All samples collected at the piezometers (SP1).

For GW$_L$, initially the EC in the mesocosms were below 200 µS cm$^{-1}$, until day 24, a period when the *Fv/Fm* of *L. lagunae* varied by up to 0.70. After this period, the *Fv/Fm* decreased and the EC, reaching values of 591 µS.cm$^{-1}$ on day 83, in the same week that the *Fv/Fm* reached zero.

A correlation of *Fv/Fm* with dissolved oxygen was also observed in the mesocosms (Spearman's $\rho$ 0.78). Figure 7b shows that the decrease in DO was accompanied by a decrease in *Fv/Fm*, and values below 5 mg L$^{-1}$ (on the 48th day of the experiment) were stressful conditions for *L. lagunae*. From this date onwards, the DO remained at an average of 2.93 ± 0.97 mg L$^{-1}$. Even with the low concentrations of DO, the absence of nutrients was the factor that most limited the recovery of *L. lagunae*, since the plants irrigated with GW$_L$* presented DO values above 3.00 mg L$^{-1}$ until the 48th day and after this period; even with the average value of 1.56 ± 0.95 mg L$^{-1}$, the *Fv/Fm* ratio remained similar to those in the other treatments.

3.1.4. Performance of the Mesocosms for the Treatment of Greywater

Figure 8 shows the average COD (a) and TS (b) loads applied to the mesocosms and removed from the liquid phase in the groups irrigated with GW$_L$ and GW$_L$* and comparing the two species. Both plants presented satisfactory results for COD and TS removal. The groups irrigated with GW$_L$* exhibited better removal. The highest applied COD loads occurred for the GW$_L$* group, mainly for *L. lagunae*, 34.22 ± 19.91 g m$^{-2}$ day$^{-1}$, while *P. acuminatum* received 11.06 ± 4.05 g m$^{-2}$ day$^{-1}$. The load for *L. lagunae* was higher because the plants that received nutrients grew more, and consequently evapotranspired more, demanding greater frequency and volume of irrigation solution.

The trend of high loadings for *L. lagunae* irrigated with GW$_L$* was also observed for TS, with values of 94.57 ± 52.99 g m$^{-2}$ day$^{-1}$, while that of *P. acuminatum* was 46.20 ± 25.58 g m$^{-2}$ day$^{-1}$. Regarding the removal, *L. lagunae* showed the best responses (Table 4), with 55% for the mesocosms irrigated with GW$_L$ and 58% for the mesocosms irrigated with GW$_L$*. The set of data used for calculations are available in Table S2 as supplementary material.

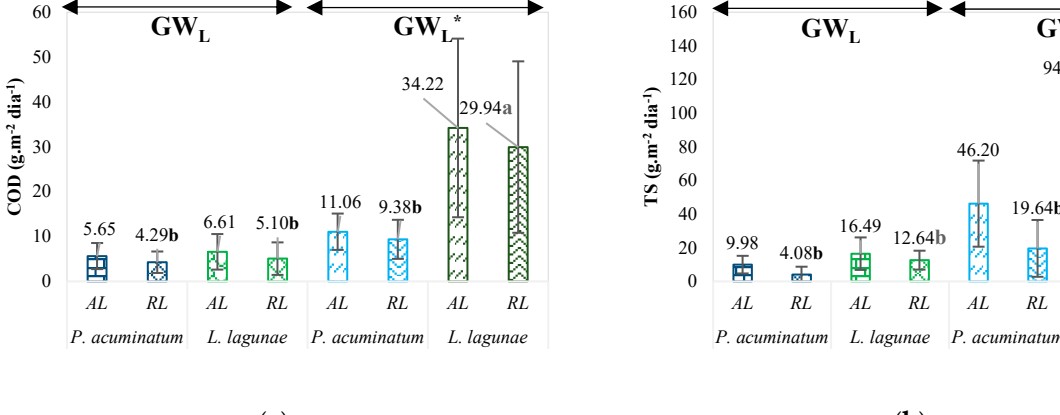

(**a**)                                                 (**b**)

**Figure 8.** Average applied and removed loads of COD and TS (g m$^{-2}$ day$^{-1}$) during 212 days of irrigation of *P. acuminatum* and *L. lagunae* with laundry greywater supplied with nutrients (GW$_L$*) and laundry greywater without nutrients (GW$_L$). AL (applied load) based on the irrigation solution concentration; and RL (removed load): results calculated from samples collected in the sampling port 2 (SP2) representing effluent samples. Lowercase letters (**a**,**b**) indicate statistical groupings based on ANOVA and Tukey tests. Values followed by the same lowercase letter are not significantly different at *p* < 0.05. Bars indicate standard errors.

**Table 4.** Average percentage removal of COD and total solids load in the mesocosms irrigated with laundry greywater with (GW$_L$*) and without (GW$_L$) nutrient supplementation.

| | GW$_L$ | | GW$_L$* | |
|---|---|---|---|---|
| | *P. acuminatum* | *L. lagunae* [a] | *P. acuminatum* | *L. lagunae* |
| COD (%) | 74.15 ± 15.38 | 75.17 ± 16.09 | 82.06 ± 18.29 | 88.64 ± 6.55 |
| TS (%) | 24.82 ± 35.56 | 55.13 ± 18.37 | 41.96 ± 24.72 | 57.77 ± 30.70 |

[a] After day 51, the mesocosms of *L. Lagunae* did not contain the aerial parts of the plants.

## 4. Discussion

The Pantanal aquatic macrophytes *P. acuminatum* and *L. lagunae* studied in the zero-liquid discharge system irrigated with greywater without nutrient supplementation did not exhibit stress during the adaptation phase. Both species showed strengths and weaknesses for use in zero-liquid discharge systems.

Throughout the 212 days of the experiment, greywater without nutrient supplementation did not offer the minimum quantity of nutrients required to maintain the growth and survival of *L. lagunae*, while *P. acuminatum* survived at the same conditions, though presenting loss of vigour over time. However, with nutrient supplementation, both species showed the best growth and performance with greywater instead of tap water.

Healthy plants with C3 photosynthesis have an *Fv/Fm* value of approximately 0.80 [27,28], and in plants in general it is possible to find values between 0.75 and 0.85 [29]. A decrease in this ratio can be attributed to processes such as photoinhibition and UV-B stress [28]. The correlation analysis showed that *P. acuminatum* was affected by the increase in temperature of the irrigation water, which decreased the ORP value and decreased the availability of dissolved oxygen in the water that is necessary for root respiration, leading to a restriction in ATP production [27]. Despite these conditions, *P. acuminatum* remained alive, although with a much lower growth rate compared to the same species in the GW nutrient-supplemented mesocosms. While the temperature of the irrigation solutions with and without nutrients was the same, the nutrient-supplied *P. acuminatum* had a greater capacity to face stress, contributing to better vigour of the plants in the greenhouse environment and providing better conditions for metabolism.

The literature indicates that for C3-type plants, the maintenance of nutritional condition promotes improvements in thermal dissipation, reduction of the effective absorption of photons, and metabolic adaptations [30]. On the other hand, C3 species may be affected by stress conditions such as high temperature [30,31], water shortage [30,32], high salt concentration [33,34] or nutritional problems [35,36]. Habermann et al. (2022) [37] identified smaller stomata and thinner mesophyll tissue in the leaves of *Stylosanthes capitata* (C3) and *Megathyrsus maximus* (C4) when exposed to elevation of temperature by 2 °C. In the current study, the production of the greywater used for irrigation was uniform, using the same washing cycle, amount of clothes, and washing products; therefore, the elevation of water temperature was related to the ambient temperature. The parameter with the greatest influence on the decrease in photosynthetic activity for *P. acuminatum* was the ORP, but one of the greatest limitations was the temperature of the irrigation solution, meaning that the local environmental condition caused stress for *P. acuminatum* irrigated with $GW_L$.

For *L. lagunae*, the main correlation of *Fv/Fm* was with electrical conductivity. The effects on EC of irrigating with greywater were observed by Al-Hamaiedeh and Bino (2010) [38], Pinto, Maheshwari, and Grewal (2010) [39], and Rodda et al. (2011) [40]. The increase in EC suggests that there was an accumulation of salts, particularly sodium chloride, over time. The accumulation of Na in the medium can hinder water uptake by the plant [40], affect plant function, and absorb and accumulate toxic ions in the plant [41]. For the mesocosms that received $GW_L$*, the salinity value was $9.9 \pm 5.5\%$ NaCl. Even with salt accumulation, the presence of other nutrients (macro and micro) may have contributed to the better vigour of *L. lagunae* even under such conditions. For *L. lagunae*, the decrease in oxygen rate may have occurred because of the accumulation of Na, as it can cause sodicity in the soil [40,41], hindering plant growth and preventing soil infiltration, drainage, and diffuse atmospheric oxygen flow [41].

It is worth noting the inverse correlation of pH of the greywater irrigation with added nutrients (TW* and $GW_L$*) of −0.65 and −0.51, respectively. Although low, this correlation may reinforce the need for the addition of nutrients. The pH for both tap and laundry greywater were approximately 12% and 15% lower with nutrient supplementation, respectively. For both irrigation solutions, there was an inverse correlation, which might indicate that pH 5 is a suitable condition *for L. lagunae*. The literature indicates that the ideal pH to maintain available essential nutrients for the plant ranges between 5 and 6 [42]. This correlation was not observed for *P. acuminatum*, which presented more stable photosynthetic activity for both irrigation solutions.

Both species were able to receive high COD loads, with an average of $34.22 \pm 19.91 \, \text{g m}^{-2} \, \text{day}^{-1}$ for *L. lagunae* and $11.06 \, \text{g m}^{-2} \, \text{day}^{-1}$ for *P. acuminatum*, with an average COD removal of approximately 85% for both. The mesocosms planted with *L. lagunae* and irrigated with $GW_L$* received a COD load higher than the range of a full-scale NBS. For instance, the recommended average load for a full-scale horizontal subsurface flow CW is $23.71 \, \text{g m}^{-2} \, \text{day}^{-1}$ [43]. Despite the higher load, the COD removal by the mesocosms was satisfactory, with removal of 87% for the $GW_L$*-treated *L. lagunae*. The COD removal was also satisfactory for the other conditions, especially when compared to the full-scale process. Literature shows that, on average, a horizontal flow CW removes up to 75% COD, and a vertical flow CW removes values above 92% [43].

The lower removal rates of total solids in comparison to COD may be related to the production of mass particles in the mesocosm, such as divalent metallic sulphides produced under anaerobic conditions [11], which are favoured by the low ORP in the mesocosms.

When comparing the two studied species, we can say that *L. lagunae* is more advantageous in terms of evapotranspiration capacity, with greater potential for use in NBS systems in urban areas to abate heat islands, because it presents a much larger number of leaves, whereas the leaves of *P. acuminatum* are thinner, longer, and fewer. The number of leaves is important for the rate of evapotranspiration and also for photosynthetic production. Evapotranspiration can provide vegetation with the role of a natural temperature regulator [44], as blocking the sun's rays and absorbing solar radiation can supply cooling [44,45]. Thus,

evapotranspiration is a relevant parameter for NBSs in local microclimates [46,47]. On the other hand, *P. acuminatum* had a more resistant metabolism and a higher photosynthetic capacity without nutrient supplementation, indicating that it is a more robust species under the regular greywater feeding system. We consider that the achieved results are promising, especially considering that the experiments were carried out under extreme conditions, such as non-nutrient supplementation and zero-liquid discharge systems where pollutants may accumulate. Experiments carried out with other species show that they adapt better, even flowering without nutrient supplementation, in open-air environments in traditional non-zero discharge systems, such as constructed wetlands [6,15].

The use of photosynthetic activity, a simple and low-cost method, proved to be very efficient in monitoring the plants and could be used during the maintenance of NBSs, for anticipating changes in the qualitative characteristics of the effluent due to the performance of the plants.

**Supplementary Materials:** The following supporting information can be downloaded at: https://www.mdpi.com/article/10.3390/resources11100084/s1, Table S1: Correlation data between Fv/Fm and physicochemical parameters for the four irrigation solutions used for L. lagunae and P. acuminatum; Table S2.

**Author Contributions:** Conceptualisation, K.T., P.P. and N.Y.; methodology, K.T., N.Y., L.L., A.C. and P.P.; validation, K.T. and P.P.; formal analysis, K.T.; investigation, K.T. and G.A.; resources, V.P., A.C. and P.P.; data curation, K.T. and G.A.; writing—original draft preparation, K.T. and P.P.; writing—review and editing, K.T., P.P., N.Y., L.L. and A.C.; visualisation, K.T.; supervision, P.P. and N.Y.; project administration, P.P.; funding acquisition, P.P. and N.Y. All authors have read and agreed to the published version of the manuscript.

**Funding:** This research was funded by: National Foundation for Health-FUNASA (project No. 25100.015.578/2017-10); Conselho Nacional de Desenvolvimento Científico e Tecnológico-CNPq; Fundação de Amparo à Pesquisa do Estado de Minas Gerais-FAPEMIG; Instituto Nacional de Ciência e Tecnologia em Estações Sustentáveis de Tratamento de Esgoto-INCT "ETEs Sustentáveis" (INCT "Sustainable Sewage Treatment Plants"); Coordenação de Aperfeiçoamento de Pessoal de Nível Superior-Brasil (CAPES)-Finance Code 001 and PROBRAL Programme-CAPES/DAAD project No. 88881.371339/2019-01). APC was funded by the Federal University of Mato Grosso do Sul-UFMS/MEC-Brazil.

**Data Availability Statement:** Not applicable.

**Conflicts of Interest:** The authors declare no conflict of interest.

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
