# Peer review of "Relationship of Photosynthetic Activity of Polygonum acuminatum and Ludwigia lagunae with Physicochemical Aspects of Greywater in a Zero-Liquid Discharge System"

_resources, doi:10.3390/resources11100084_

Round 1

Reviewer 1 Report

1. In the abstract, the use of abbreviations of scientific names should be consistent after the first full mention . (Page 1, Line 25) 2. The author stated that preliminary study was done using 3 species and needed further investigation, However, only 2 species from the preliminary study were chosen out of 3. Author needs to clarify why the selected 2 species were chosen. What is the difference and what is the significance of those species to the current study? (Page 2, line 68) 3. It is suggested for the author to provide specific composition of modified Hoagland solution which was used as nutrient supplementation in this study. (Page 3, line 86) 4. Author needs to clarify and clearly explain why cold water, only jeans type and cotton program were chosen in this study?  Any significant reasons for choosing them for the production/ quality of greywater? (Page 3, line 88) 5. The caption for table 1 is confusing. Author should recheck and correct. (Page 4, line 117) 6. It is suggested to add reference to the statement in page 4, line 128 7. It is suggested for the author to add reference to the experimental procedure which they are referring to. (Page 4, line 133).  8. There is an error found in the manuscript. Author should recheck (Page 7, line 208) 9. It is suggested to the author to add any supported reference to the statement in page 8, line 231. 10. Figure 8 in this manuscript cannot be found. Author should recheck. (Page 11, line 304) 11. Author should also explain the survival and growth of Pacuminatum without supplementation of nutrients in greywater. (Page 11, line 332). 12. Some of the references need to be corrected. Author should recheck (Page 13,line 435)

Reviewer 2 Report

Dear Authors,

Manuscript No. resources-1903746 is about the possibility of irrigation the ornamental plants using grey water from laundry. Nowadays the topic of zero-waste and zero-discharged systems is very important. In the manuscript, some aspects are not taken up and some elements require improvement or clarification. Below are my detailed comments.

1) Lines 53-61: Is the main idea that ornamental plants are more to be expected on urban cooling islands? The main message is not clear to other than landscape architects. Please highlight this.

2) Line 57: This sentence is surprising to me. Knowledge of phytoremediation has a long tradition and is well documented (line 43).

3) Please specify the supplementation in detail – mineral composition and dose of the added nutrients. It is mandatory because the concentration of nutrients in irrigation samples wasn't measured. If the nutrients were measured please add the results.

4) The results will be more usable for water/wastewater engineers when the parameters of inlet and outlet for all research series will present. But not the average values or median, just real values during all time of the experiment. The COD results in the outlet stream are important if the stream needs to be managed.

5) Why do you assume there were no nutrients in the laundry wastewater? You are using a simplification that is not correct: no addition of Hoagland solution = no nutrients. However, phosphates can occur in laundry wastewater, especially when you used the softener (perhaps sodium orthophosphate). So some nutrients can be present in laundry wastewater, but not all that were in demand. Please do a more precise description. lines 231-234, 244-250

6) In the presented experiment the series with ponds containing gravel and no plants and fed by four types of irrigation water are missing. There are no comparative series to evaluate the influence of biochemical processes taking place in the porous material. Has the development of a biological membrane been noted? It has not been documented that plants had the only effect on COD decrease.

7) Figure 6. Decreasing ORP following decreasing dissolved oxygen and increasing T could be the result of the activity of anaerobic microflora generated on the gravel. A series of results with COD is needed.

8) Lines 298-301: The results of GWL* similar to GWL aren’t included.

9) Table 3. The standard deviation of EC results in TW* and GWL* series is incredibly high! Does such variability in the initial sample allow for meaningful correlation analysis?

10) Table 3. Was tap water really so low-mineralized or it is a mistake?

11) Figure 6. Are these the measurements of the outlet streams from ponds? It isn’t specified.

12) How long was the acclimatisation period?

13) How long was the experiment? I find 212 and 215 days duration (lines 210, 252, the caption of Fig.5)

14) Fig. 5 – only the results during 170 days are shown however, the caption suggests 212 days

15) Line 161: TDS is total dissolved solids, not salts

16) Just EC results are enough. EC is in strict correlation with the salinity and with TDS. The excessive results make comprehension difficult and don't make the interpretation easier (Fig 7a and 7c).

17) In Figure 7 the ORP values are missing.

18) Table 3 – the number of samples in superscript looks like the exponent. The first impression is unfavourable, you think there is a mistake. Maybe the number in parenthesis will be better.

19) line 164: Firstly I wrote 2,200 mL (2,2 L) and after considering the volume of pots and the porosity of gravel I read it correctly as 200 mL J

20) Figure 8 is missing.

21) Editing errors in lines 208, 217-219, 236-238

 Best regards

Round 2

Reviewer 2 Report

1) The Authors made some corrections in the text. However, the change tracking mode was applied incorrectly (all changes are crossed out), so I am not able to confirm that the proposed changes are appropriate.

2) The doses of applied nutrients are still unknown. This information is obligatory to show to the readers.

3) Table S1: In the manuscript are EC and DO abbreviations and CE and OD in the table. In the top lines of this table, the different language is applied.

4) Is Figure 6a-d  presented the results of measurements in irrigation water or water from a piezometer (after 7 days of stagnation)? It is unclear and it does not follow the caption. The same is Fig. 7.

5) What does the mark * next to the value -0.39 in table S1 mean?

6) Sending readers to ask questions by e-mail is not a good solution for a research paper. If you don’t want to present all data of influent and water from ponds, please consider preparing a table similar to Table 3, but containing the summary of parameters of stagnant water (treated???).

7) In the arrangement of the experiment and the discussion of the results, an extensive analysis of plant parameters is carried out. Unfortunately, too little attention has been paid to the properties and treatment of wastewater, especially the effluent streams. After all, on plant islands, these effluent streams should be properly managed if they do not meet the release requirements. Plant communities are commonly used for wastewater treatment. However, such an analysis is missing from the manuscript but is very important for the application of these solutions in urban practice.

Author Response

Dear reviewer,

Thanks for your new comments, we hope that our answers and corrections in the manuscript are accordingly. We are open to discuss anything you feel it is not sufficient.

1) The Authors made some corrections in the text. However, the change tracking mode was applied incorrectly (all changes are crossed out), so I am not able to confirm that the proposed changes are appropriate.

Answer: We are sorry for this problem with track & changes. We took a look at the pdf generated by the editorial office system and it is really very confusing! When sending the revised version, we warned the editor’s office in the covering letter. During the first submission, some errors were generated when editorial office system converted the word document into pdf. It seems that they don’t check the pdf file before releasing to the reviewers. Trying to overcome this problem, we attached a pdf built by ourselves, so that you can follow all the changes made in the manuscript.

2) The doses of applied nutrients are still unknown. This information is obligatory to show to the readers.

Answer: You are right, we added the concentrations used for the stock solutions but not the volume of the stock solutions used for irrigation water volume… We fixed it, complementing the information, so that the reader can reproduce the experiment if desired. Now it reads:

“Eleven 11 stock solutions (SS) were prepared and different volumes were added to the irrigation solution when required. The volume added of each solution and their chemical composition (g L-1) were as follows: SS-A (5 mL L-1) Ca(NO3)2.4H2O: 236; SS-B 5 mL L-1) KNO3 : 101; SS-C 2 mL L-1) MgSO4.7H2O: 246.5; SS-D (1 mL L-1) KH2PO4: 136; SS-E (1 mL L-1) EDTA: 13.0; SS-F (1 mL L-1) FeCl3.6H2O: 7.80; SS-G (1 mL L-1) MnCl2.4H2O: 1.810; SS-H (1 mL L-1) H3BO3: 2.860; SS-I (1 mL L-1) ZnSO4.7H2O: 0.220; SS-J (1 mL L-1) CuSO4.5H2O:  0.080 and; SS-K (1 mL L-1) H-2MoO4: 0.020.”

3) Table S1: In the manuscript are EC and DO abbreviations and CE and OD in the table. In the top lines of this table, the different language is applied.

Answer: Checked and corrected.

4) Is Figure 6a-d  presented the results of measurements in irrigation water or water from a piezometer (after 7 days of stagnation)? It is unclear and it does not follow the caption. The same is Fig. 7.

Answer: In Figure 6 this information is in the figure caption. It was added in the previous corrected version …“Samples collected at the piezometers (SP1) except for (b) where; I: irrigation solution”… To complement we added the same information in the caption of Figure 7.

5) What does the mark * next to the value -0.39 in table S1 mean?

Answer: The mark * was left there by mistake. It is corrected now.

6) Sending readers to ask questions by e-mail is not a good solution for a research paper. If you don’t want to present all data of influent and water from ponds, please consider preparing a table similar to Table 3, but containing the summary of parameters of stagnant water (treated???).

Answer: We don’t have problems with presenting the data, we just felt like unnecessary. We included a excel file with the raw data as requested. We included in the text the following sentence (lines 34-341) “The set of data used for calculations are available in Table S2 as supplementary material.”

7) In the arrangement of the experiment and the discussion of the results, an extensive analysis of plant parameters is carried out. Unfortunately, too little attention has been paid to the properties and treatment of wastewater, especially the effluent streams. After all, on plant islands, these effluent streams should be properly managed if they do not meet the release requirements. Plant communities are commonly used for wastewater treatment. However, such an analysis is missing from the manuscript but is very important for the application of these solutions in urban practice.

Answer: We are not sure we understand in which sense this discussion is missing. Even though there is no effluent production (zero-liquid discharge system) we presented and discussed the performance of the mesocosms irrigated with greywater regarding COD and total solids removal, in percentage and loading rate (item 3.1.4 and lines 403 to 416).  
